# Evaluation of Antifouling Potential and Ecotoxicity of Secondary Metabolites Derived from Red Algae of the Genus *Laurencia*

**DOI:** 10.3390/md17110646

**Published:** 2019-11-16

**Authors:** Maria Protopapa, Manto Kotsiri, Sofoklis Mouratidis, Vassilios Roussis, Efstathia Ioannou, Skarlatos G. Dedos

**Affiliations:** 1Department of Biology, National and Kapodistrian University of Athens, Panepistimiopolis Zografou, 15784 Athens, Greece; mariaprot@hotmail.com (M.P.); madokot@yahoo.gr (M.K.); aesofos@gmail.com (S.M.); 2Section of Pharmacognosy and Chemistry of Natural Products, Department of Pharmacy, National and Kapodistrian University of Athens, Panepistimiopolis Zografou, 15771 Athens, Greece; roussis@pharm.uoa.gr

**Keywords:** *Laurencia*, red algae, biofouling, ecotoxicity, *Amphibalanus amphitrite*, barnacles, perforenol

## Abstract

Red algae of the genus *Laurencia* are known to biosynthesize and secrete an immense variety of secondary metabolites possessing a spectrum of biological activities against bacteria, invertebrates and mammalian cell lines. Following a rigorous cross-species screening process, herein we report the antifouling potential of 25 secondary metabolites derived from species of the genus *Laurencia*, as well as the thorough evaluation of the ecotoxicity of selected metabolites against non-target marine arthropods and vertebrate cell lines. A number of these secondary metabolites exhibited potent antifouling activity and performed well in all screening tests. Our results show that perforenol (**9**) possesses similar antifouling activity with that already described for bromosphaerol, which is used herein as a benchmark.

## 1. Introduction

Owing to their co-evolution with living organisms and the environment, natural products offer increased specificity and efficiency towards their targets as a means of increasing the fitness and survival of the organism which produces them [1,2,3]. Marine organisms are no exception to this rule, producing an astonishing array of natural products with diverse biological activities [1,2,3]. A characteristic example in the marine environment is the *Laurencia* paradox [4], referring to species of red algae belonging to the genus *Laurencia* that biosynthesize a constantly increasing number of new secondary metabolites, which to date exceeds 1,000 chemical entities, with structural and functional diversity [5]. Red algae of the genus *Laurencia* have been a rich source of structurally diverse and unique secondary metabolites with both attractant and deterring activity against a variety of marine organisms [4,5], which in turn has triggered the interest of researchers towards the evaluation of the isolated metabolites as naturally-occurring chemical compounds with antifouling activity against marine fouling invertebrates.

Although the biosynthetic mechanisms of their secondary metabolites are incompletely understood and proposed to be mediated by haloperoxidases [6] and lactoperoxidases [7] due to the presence of high proportions of brominated, chlorinated and lactone-containing metabolites [4,5,8], their chloroplast is proposed to play a role in the synthesis of their secondary metabolites [6]. Structures within the cells of red algae, mostly localized close to or at the surface cell layer [6], act as storage vesicles for secondary metabolites that help these cells avoid autotoxicity [9] from the metabolites they produce. These subcellular structures can occupy the whole cell, such as in the case of gland or vesicle cells [10], or can be refractive vesicles, such as the *corps en cerise* [6,11] and the physodes [12]. Irrespective of the intracellular structural form, the transport and exocytosis of the secondary metabolites is much better understood [13] than their biosynthetic pathways [4,8], being shown to occur via actin microfilaments and microtubule-mediated exocytosis [6]. Membranous tubular connections [6] or stalk-like structure connections [14] have been proposed to assist transport of the vesicle content to the thallus surface and thus elicit chemical defense against marine bacteria [14,15] or elicit chemical attraction for the herbivorous marine gastropod mollusk *Aplysia brasiliana* [16]. Despite the fact that results from in-vitro experiments have shown that the thallus surface concentrations of secondary metabolites, such as elatol [17,18], are low enough to elicit a chemical defense against fouling organisms, the presence of bromine and chlorine in subcellular vesicles is evidence compelling enough to suggest that an increase in fouling pressure and programmed cell death events can cause exudation of metabolites and concomitantly elicit chemical defense [6].

Understanding the distribution of surface-active molecules of red algae in situ and their chemical, biological and ecological consequences have been the pursuit of several studies [19], where the focus has been either on the behaviour of sea hares of the genus *Aplysia* feeding on *Laurencia* red algae and the subsequent use of the *Laurencia* secondary metabolites by the sea hares [16], or the documented settlement deterrence activity of *Laurencia* secondary metabolites against barnacle species or other marine invertebrates and bacteria [18,19,20].

Barnacle species, and especially the cyprids of the model organism *Amphibalanus amphitrite*, due to the already established protocols for settlement bioassays [21,22,23], have become one of the most frequently used animals to evaluate the antifouling activity of isolated secondary metabolites or synthetic compounds [18,20,23,24,25,26]. To establish a gregarious mode of settlement, *A. amphitrite* cyprids deposit a protein named Settlement Inducing Protein Complex (SIPC) which acts as a pheromone cue for their conspecifics to settle nearby and sexually reproduce [27,28,29,30,31,32].

Besides the use of *A. amphitrite* cyprids, as a model fouling invertebrate in laboratory bioassays of the efficacy of new antifouling formulations, other key biofouling organisms, such as bacteria, microalgae, fungi, macroalgae and other invertebrates, have also been used [17,33,34,35,36]. Providing thorough and rigorous evaluation of the antifouling potential and ecotoxicity of natural compounds has always been the critical point in any documented endeavor to identify new and useful chemical compounds [37]. Not without their limitations [17], such laboratory bioassays offer versatility and ease of assessment when several different compounds have to be tested against a variety of organisms. Several authoritative reviews [33,37,38] have proposed selection and evaluation criteria in the screening of potential antifouling compounds, such as the therapeutic ratio of > 50 [38]. However, the absence of any concrete knowledge on the mode of action of antifouling compounds against biofouling organisms [39] makes it difficult to put forth a single molecule as satisfying all the antifouling and ecotoxicity criteria that have been proposed.

With these limitations in mind and adhering to already proposed evaluation criteria for antifouling compounds [37,38], we screened 25 secondary metabolites isolated from *Laurencia* species, as well as *Aplysia* species feeding on the red algae, aiming to assess both their antifouling potential and their ecotoxicity profile. We followed a strict elimination protocol that relied first on the assessment of the settlement inhibitory activity of these metabolites against cyprids of *A. amphitrite*, followed by the evaluation of their toxicity against *Artemia salina*, *Chaetoceros gracilis*, a trout and a human cell line, as well as species of fouling marine bacteria and concluding with the assessment of the surface exploration behaviour by the cyprids of the shortlisted secondary metabolites.

## 2. Results

Given that a number of secondary metabolites from *Laurencia* species have already been identified as possessing antifouling activity, we designed a rigorous progressive testing procedure in which a number of compounds isolated from *Laurencia* and *Aplysia* species were tested and progressively eliminated in a successive series of model organisms to assess their potential use as antifouling compounds, as well as their ecotoxicity profile.

### 2.1. Evaluation of Settlement Inhibition of Amphibalanus amphitrite

Initially, we evaluated 25 secondary metabolites isolated from species of the genus *Laurencia*, as well as sea hares of the genus *Aplysia* feeding on them (**1**–**25**, Figure 1) in settlement bioassays with the model barnacle species *A. amphitrite*. The cumulative results are shown in Table 1 and Appendix A.

Several metabolites displayed EC_50_ values for settlement inhibition in the lower micromolar range. Some of them, such as laurinterol (**6**), 4-hydroxy-5-brasilene (**8**) and elatol (**12**) had an obvious toxic effect on cyprids (Table 1 and Appendix A), while obtusallene I (**24**) and chondrioallene (**25**) had a settlement-stimulating activity at the higher concentrations tested (Table 1 and Appendix A). Settlement to no metamorphosis ratio at 72 h (Table 1) was calculated to identify whether any metabolite had an undesirable developmental effect on the cyprids [23], with filiformin (**5**) and (3*E*)-laurenyne (**21**) scoring very high on this test. Bromosphaerol, a brominated diterpene (Figure 1) isolated from the red alga *Sphaerococcus coronopifolius* [24] and used in the current study as a positive control, showed a similar settlement-inhibiting profile, as previously reported [24], but its presence also resulted in a large percentage of animals not progressing to metamorphosis (Table 1 and Appendix A).

With bromosphaerol serving as a benchmark, eight compounds (**4**, **9**, **11**, **12**, **14**, **18**, **19,** and **22**) were satisfying the criterion of consistently displaying sub-micromolar EC_50_ values for settlement inhibition of barnacle cyprids (Table 1).

### 2.2. Evaluation of Toxicity Against Artemia salina

The eight secondary metabolites that were evaluated as having potentially promising antifouling activity against *A. amphitrite* (Table 1), as well as bromosphaerol, were subsequently tested in *A. salina* nauplii toxicity assays at 24 and 48 h of exposure (Table 2). These toxicity assays were conducted to identify if any of the selected metabolites exhibit toxicity towards another invertebrate organism and eliminate those that confer toxicity after longer exposure times (i.e. at 48 h). The exclusion criterion was a >10-fold increase in the EC_50_ value between the 24 h and 48 h time points. The results showed that iso-laurenisol (**4**) and elatol (**12**) did not pass this criterion and were therefore eliminated from further testing. Bromosphaerol did not show any toxicity against *A. salina* nauplii (Table 2 and Appendix A).

### 2.3. Evaluation of Toxicity against Chaetoceros gracilis

For phytoplankton growth toxicity assays, the criterion for assay validity was that the cell count of the negative control after 96 h should be ≥2 × 10^5^ cells/mL [40]. The exclusion criterion was that a metabolite should exhibit a lower EC_50_ value than bromosphaerol, which proved to be a very potent inhibitor of *C. gracilis* growth. The results showed that perforatone (**11**), 3,15-dibromo-7,16-dihydroxy-isopimar-9(11)-ene (**14**) and laurencienyne (**22**) did not pass this criterion and were therefore eliminated from further testing (Table 2 and Appendix A).

### 2.4. Evaluation of Cytotoxicity Against RTL-W1 Cell Line

The RTL-W1 cell line is a non-transformed rainbow trout liver cell line established from the normal liver of an adult rainbow trout by proteolytic dissociation of liver fragments [41]. Shortlisting three metabolites, namely perforenol (**9**), the dactylomelane diterpene **18** and neorogioldiol (**19**), we employed three different cytotoxicity tests in an attempt to identify possible modes of action through which each secondary metabolite is affecting cell viability, a prerequisite for any downstream application of these compounds [39]. The results showed (Table 2 and Appendix A) that none of the three tested metabolites had an effect on lysosomal activity and/or the general metabolism and mitochondrial activity of this trout liver cell line. The crystal violet assay showed that two of the tested metabolites had an EC_50_ value at the lower micromolar concentrations (Table 2), while compound **18** and bromosphaerol had sub-micromolar EC_50_ values (Table 2). Since none of these secondary metabolites proved to be more toxic than bromosphaerol or had an explicit toxic profile in all three assays, none was excluded from subsequent tests.

### 2.5. Evaluation of Cytotoxicity against HEK293 Cell Line

The HEK293 cell line is a transformed human embryonic kidney cell line of epithelial origin [42]. Testing the same three metabolites (**9**, **18** and **19**) as in the RTL-W1 cell line, using three different cytotoxicity tests, we observed (Table 2 and Appendix A) that in the case of HEK293 cells, metabolite **18** and neorogioldiol (**19**) were very selective against mitochondrial function and metabolism, unlike the RTL-W1 cell line. On the contrary, lysosomal activity was not affected by the presence of any of the three tested metabolites (Table 2). The crystal violet assay (Table 2) showed that perforenol (**9**) and bromosphaerol exhibited mild cytotoxicity.

### 2.6. Evaluation of Growth of Marine Bacteria

The activity of perforenol (**9**), the dactylomelane diterpene (**18)** and neorogioldiol (**19**), as well as bromosphaerol was evaluated against five strains of marine bacteria known to exist in biofilms, namely, *Bacillus subtilis, Bacillus mojavensis, Bacillus thuringiensis, Pseudomonas putita* and *Pseudomonas pseudoalcaligenes*. The results (Appendix A) showed that none of these metabolites had any effect on the growth of the tested bacterial strains.

### 2.7. Effect of Perforenol and Bromosphaerol on the Settlement Exploration Behaviour of Amphibalanus amphitrite

Finally, in order to assess whether perforenol (**9**) and bromosphaerol had any effect or interfered with the settlement behaviour of cyprids of *A. amphitrite*, we carried out modified Western blot assays upon completion of the settlement bioassays with the barnacle cyprids. The results (Figure 2) showed that concentrations of perforenol (**9**) and bromosphaerol at the EC_50_ values had no statistically significant effect (*p* > 0.05) on the deposition of SIPC protein on the surface of the bioassay plates. Bromosphaerol, even at a concentration of 10 μM, did not affect deposition of SIPC by the cyprids relative to the control (ASW) (Figure 2).

## 3. Discussion

Although the concept of identifying marine natural products as antifouling agents is not new, reports on the variety of secondary metabolites that have antifouling potential against barnacle species has been increasing recently [33,34,37,39,43,44]. Research on the bioactivity and ecological functions of secondary metabolites from red algae of the genus *Laurencia* has been exponentially increasing in the last two decades, with more than 1,000 compounds added in an ever-increasing list of isolated metabolites [4,5].

In the framework of our ongoing interests in the discovery of bioactive metabolites from marine organisms of the East Mediterranean Sea, we screened 25 compounds isolated from *Laurencia* and *Aplysia* species, including 13 sesquiterpenes (**1**–**13**), 7 diterpenes (**14**–**20**) and 5 C_15_ acetogenins (**21**–**25**), aiming to assess both their antifouling potential and their ecotoxicity profile. Initially, we evaluated the settlement inhibitory activity of these metabolites against cyprids of the barnacle *A. amphitrite*. By benchmarking the antifouling potential of these metabolites against the already established antifouling activity of bromosphaerol isolated from the red alga *Sphaerococcus coronopifolius* [24], we identified several of them as having equal or better antifouling activity than bromosphaerol (Table 1). The eight most potent metabolites (**4**, **9**, **11**, **12**, **14**, **18**, **19,** and **22**) were sequentially evaluated for their toxicity against *A. salina*, *C. gracilis*, a trout and a human cell line, as well as five species of marine fouling bacteria (Table 2), attempting to exclude any possessing toxicity against other marine non-target organisms that would render these metabolites unsuitable for antifouling applications. Among these, perforenol (**9**), originally isolated from *Laurencia perforata* and *Laurencia chondrioides* [45,46], displayed potent antifouling activity and minimal ecotoxicity.

An extensive literature search revealed that among *Laurencia*-derived metabolites, only omaezallene [20] and 2,10-dibromo-3-chloro-7-chamigrene [26] have exhibited substantially low EC_50_ values and deserve being considered for antifouling formulations. Herein, we showed that perforenol (**9**) can also be included in this short list of secondary metabolites with potent antifouling activity.

Since several secondary metabolites deriving from *Laurencia* species possess antifouling activity against the barnacle *A. amphitrite*, we propose that the ability of *Laurencia* species to deter fouling derives from the concurrent presence of several metabolites, while any antifouling application of these metabolites does not have to rely on the exclusive use of each one of them.

## 4. Materials and Methods

### 4.1. Extraction and Isolation of Metabolites

Compounds **1**–**3**, **5**, **16**–**18** and **23** were isolated from *Aplysia depilans* Gmelin, 1791 (Gastropoda: Heterobranchia: Aplysiida: Aplysiidae [47]), collected off Skyros Island, Greece, at a depth of 2–4 m, in August of 2011, as described by Petraki (2016) and Petraki et al. (2015) [48,49]. Compounds **4** and **13** were isolated from *Laurencia microcladia*, collected at the coastline of Ag. Kyriaki, in Tinos island, at a depth of 0.5–2 m, in September of 2011, while compounds **4**, **24** and **25** were isolated from *Laurencia obtusa*, collected at the coastline of Ag. Sostis, in Tinos island, at a depth of 0.5–2 m, in September of 2011, as described by Harizani (2013) [50]. Compounds **6**–**8**, **19**, **20** and **22** were isolated from *Laurencia glandulifera*, collected at Vatsa bay, in Kefalonia island, at a depth of 0.5–2 m, in May of 2014, as described by Konidaris (2015) [51]. Compounds **9**–**11**, **13** and **14** were isolated from *Laurencia* sp., collected at the gulf of Preveza, at a depth of 2–3 m, in July of 2010, as described by Giannaropoulou (2013) [52]. Compound **15** was isolated from *Aplysia fasciata*, collected in the Alfacs Bay, Delta de l’Ebre, Tarragona, Spain, at a depth of 1–1.5 m, in January of 2008, as described by Ioannou et al. (2009) [53]. Compound **21** was isolated from *Laurencia chondrioides*, collected at Ag. Theodoroi, in Kefalonia island, Greece, at a depth of 1–2 m, in August of 2010, as described by Kokkotou et al. (2014) [46]. Metabolites **1**–**25** were identified as laurene (**1**), debromoallolaurinterol acetate (**2**), allolaurinterol acetate (**3**), *iso*-laurenisol (**4**), filiformin (**5**), laurinterol (**6**), epibrasilenol (**7**), 4-hydroxy-5-brasilene (**8**), perforenol (**9**), perforenone A (**10**), perforatone (**11**), elatol (**12**), obtusenol (**13**), 3,15-dibromo-7,16-dihydroxy-isopimar-9(11)-ene (**14**), deoxyparguerol 16-acetate (**15**), three previously reported dactylomelane diterpenes (**16**–**18**), neorogioldiol (**19**), *O*^11^,15-cyclo-14-bromo-14,15-dihydrorogiol-3,11-diol (**20**), (3*Ε*)-laurenyne (**21**), laurencienyne (**22**), graciosallene (**23**), obtusallene Ι (**24**), and chondrioallene (**25**) by comparison of their spectroscopic and physical characteristics with those reported in the literature.

### 4.2. Rearing of Barnacles

Adult *A. amphitrite* (*Cirripedia*, Balanidae) were collected from boat docks at Port Mikrolimano and Floisvos, Athens, Greece. Animals were cleaned of epibionts with a small hard brush and meticulously identified as individuals of the species. Adult barnacles were kept in separate aerated glass tanks (20 L) containing 200-μm filtered natural seawater at 27 °C and a 12:12 L:D photoperiod. Tanks were fed with 24-h hatched *Artemia* sp. (*Branchiopoda*) nauplii and *Tetraselmis suecica* (*Chlorodendrophyceae)* and *Skeletonema costatum* (*Bacillariophyceae*) algae each day and seawater was changed on alternate days. Upon stress to induce oviposition (exposure to air for 24 h or immersion in fresh water for 5 h), adults were returned to seawater for larval release. Hatched nauplii were attracted to a point light source, collected and placed into a beaker containing 2 or 3 L of 0.7 μm GF/F (GE Healthcare, Little Chalfont, UK )-filtered natural seawater at a density of approximately 1–2 nauplii/mL with gentle aeration. Nauplii were maintained at 27 °C at a 12:12 L:D photoperiod on a diet of *C. gracilis* (*Bacillariophyceae*) provided at a density of 2 × 10^5^ cells/mL. Cultured in these conditions, nauplii metamorphosed to cyprids in 5 days. Aliquots of these cyprids were collected with a wide-mouthed Pasteur pipette and aged at 4 °C for 1 day prior to use in settlement assays. Only batches of cyprids that were active and had numerous oil cells, representing energy reserves, were used in settlement assays.

### 4.3. Amphibalanus amphitrite Settlement Bioassay

Assays were conducted by adding 10 cyprids into individual wells of a 24-well polystyrene sterile microplate (Orange Scientific, Braine l’Alleud, Belgium) with 2 mL of artificial sea water (ASW) (25‰) and various concentrations of the chemical compounds as listed in Table 1. These experiments were repeated at least three times with four replicates for each concentration of any given compound (*n* = 120 cyprids/compound). Stock solutions for all compounds were prepared in methanol (Scharlab, Barcelona, Spain) and serial dilutions were prepared in 0.7 μm (GF/F, GE Healthcare, Little Chalfont, UK), filtered sea water to have a final concentration of 2.5% methanol in each assay solution. Plates were covered and sealed with Parafilm^®^ to avoid evaporation and incubated at 25 °C away from any light source and examined after 24, 48 and 72 h. Each animal was inspected under a stereomicroscope and its condition was recorded. On each day, cyprids that did not move, had extended thoracopods and did not respond after a light touch with a Pasteur pipette, were regarded as dead. Both permanently attached and metamorphosed individuals were counted as settled or juveniles. The remainder were counted as ‘no metamorphosis’, i.e. animals that responded to light touch and were still cyprids at the end of the experiments with no overt signs of metamorphosis. Results were expressed as a percentage of animals that settled (settlement) or were inactive upon a light touch with a Pasteur pipette (mortality) or did not undergo settlement and metamorphosis (no metamorphosis) from the total number of animals placed in a well of a 24-well tissue culture plate. The IC_50_ or EC_50_ was determined as the concentration of each compound that resulted in 50% inhibition of the cyprids’ settlement or metamorphosis, and LC_50_ was determined as the concentration of each compound that resulted in 50% mortality of the cyprids.

### 4.4. Artemia salina Toxicity Assay

The short-term toxicity assay with *A. salina* nauplii [54,55] was used to determine the toxicity of secondary metabolites. Hatching of *A. salina* nauplii was carried out in ASW (35‰) using 0.5 g of cysts incubated in 500 mL seawater at a temperature of 25 ± 1 °C and lateral illumination (1000 Lux). All cysts were kept in continuous suspension with aeration and 24 h later hatched larvae (instar I) were harvested. Toxicity assays were carried out in three independent experiments in 24-well plates with four replicates of each tested concentration. Ten nauplii per well were transferred with a Pasteur pipette into the 24-well plates and each well was filled with 2 mL of ASW and 50 μL of various concentrations of the tested compounds in methanol. Control treatments received 50 μL of methanol. Plates were incubated in the dark at 25 ± 1 °C for 48 h. Mortality was assessed by recording the number of individuals with no movement of their appendages within 10 seconds and is expressed as the percentage of the total number of assessed individuals for each concentration.

### 4.5. Chaetoceros gracilis Toxicity Assay

The centric marine diatom *C. gracilis* Schütt (strain CCMP1315) was grown photoautotrophically in autoclaved artificial seawater (sea salts; Sigma-Aldrich, St. Louis, MO, USA) (30‰) supplemented with F/2 (half-strength medium *f* [56] as subsequently modified [57]) in 100 mL medium at 20 °C and with ambient air bubbled in the media. By monitoring cell density, an adjusted culture of >1 × 10^6^ cells/mL was used for the toxicity assays. At 96 h, a 0.9 mL aliquot of each treatment was sampled, mixed with 0.1 mL Lugol and cell density was counted. Experiments were assumed valid if cell count in the control after 96 h was ≥2 × 10^6^ cells/mL.

### 4.6. RTL-W1 Cell Line Cytotoxicity Assay

The RTL-W1 cell line, a non-transformed rainbow trout liver cell line that has already been used in environmental toxicity assays [58], was established from the normal liver of an adult rainbow trout by proteolytic dissociation of liver fragments [41]. RTL-W1 cells were cultured in 75 cm^2^ culture flasks (Orange Scientific, Braine l’Alleud, Belgium) at 20 °C in Leibovitz’s (L-15, Sigma-Aldrich, St. Louis, MO, USA) medium supplemented with 5% fetal bovine serum (FBS) and 1% L-glutamine in CO_2_-free incubator. For the cytotoxicity assays of the selected secondary metabolites, cells were seeded at 3 × 10^4^ cells/well density in a 96-well tissue culture plate (Orange Scientific Braine l’Alleud, Belgium) in 100 μL culture medium/well for 2 days before the assays. Three different cytotoxicity protocols were used: 1) The alamar blue/resazurin fluorometric/colorimetric cell viability assay [59] to assess the metabolic capacity and mitochondrial function; 2) the crystal violet cell viability assay for indirect quantification of cell death [60]; and 3) the neutral red cell proliferation assay for lysosomal activity [61]. Two days after seeding the cells, cultures were incubated with various concentrations of selected secondary metabolites at 20 °C for 48 h before cell viability was assessed. Final solvent (methanol) concentration per well in all experiments was 2.5% *v*/*v*, a concentration that was found to be non-toxic for the cells. For the alamar blue/resazurin fluorometric/colorimetric cell viability assay [59], 100 μL of 3 μg/100 μL Resazurin sodium salt (Sigma-Aldrich, St. Louis, MO, USA) in incubation medium was added on each well and incubated under sterile conditions at 20 °C for 4 h. For the crystal violet cell viability assay [60], after 48 h of incubation with each secondary metabolite, the culture medium was aspirated and cells were washed twice with sterile dd. H_2_O. The final wash was aspirated and 50 μL of 0.5% (*w*/*v*) crystal violet staining solution (20% methanol in dd. H_2_O) was added to each well followed by incubation for 20 min at room temperature on a bench rocker with a frequency of 20 oscillations per minute. The staining solution was then aspirated and the plate was washed four times with sterile dd. H_2_O and left to air-dry without its lid for 3 h at room temperature. Next, 200 μL of methanol was added to each well, and the plate was incubated with its lid on for 20 min at room temperature on a bench rocker with a frequency of 20 oscillations per minute. For the neutral red cell proliferation assay [61], after 48 h of incubation with each secondary metabolite, the culture medium was aspirated and cells were washed twice with phosphate-buffered saline (PBS, pH 7.4). Cells were then incubated with 100 μL of a 40 μg/mL neutral red staining solution in culture medium at 20 °C for 4 h followed by two washes with 150 μL of PBS. Then cells were incubated with 100 μL neutral red destain solution (50% ethanol, 49% dd. H_2_O, 1% glacial acetic acid) for 10 min on a bench rocker with a frequency of 80 oscillations per minute. Absorbance was read at 570 nm for alamar blue/resazurin and crystal violet and at 540 nm for neutral red using a Tecan infinite M200 Pro microplate reader. For each assay and each secondary metabolite, a minimum of four independent dose–response experiments with three replicates each were carried out. Calculations were carried out using detailed formulas (AbCam, Cambridge, UK, and results were analyzed with GraphPad Prism v.6.

### 4.7. HEK 293 Cell Line Cytotoxicity Assay

HEK293 cells, a human embryotic kidney cell line, were obtained from ATCC (ATCC^®^ CRL-1573^™^) at passage 37 of the original clone and used at passage 38. Cells were cultured routinely in 75 cm^2^ culture flasks (Orange Scientific, Braine l’Alleud, Belgium) at 37 °C (5% CO_2_) in DMEM cell culture medium (Biosera) supplemented with 10% FBS (Sigma-Aldrich, St. Louis, MO, USA) and 1% L-glutamine. All assay conditions and protocols were identical to those described above for RTL-W1 cells with the exception that incubations were carried out at 37 °C.

### 4.8. Marine Bacteria Growth Assay

The ability of selected metabolites to inhibit the growth of bacteria was assessed using five strains of marine bacteria involved in marine surface biofilm formation, namely, *B. subtilis, B. mojavensis, B. thuringiensis, P. putita* and *P. pseudoalcaligenes*. Bacterial strains were maintained on Zobel marine agar plates [62]. Sterile filter paper discs (4 mm) (GF/F; GE Healthcare, Little Chalfont, UK) were loaded with 40 μL samples of serial dilutions of each tested metabolite, allowed to dry at room temperature and then placed on agar plates (Zobel marine agar, pH = 7.3), which were seeded with a single strain of bacteria. Plates were incubated for 24 h at 25 °C. Marine bacteria growth inhibition was determined by measuring the zone of inhibition in mm around the filter paper disc. As a positive and negative control, standard discs were loaded with 40 μL of 0.5 M penicillin G (2 to 10 mm diameter) or solvent (40 μL methanol), respectively.

### 4.9. Substrate Exploration Analysis by Amphibalanus amphitrite Cyprids

To test the substrate exploratory behaviour of *A. amphitrite* cyprids in the presence of selected metabolites, settlement bioassays were conducted by adding 10 cyprids into individual wells of a 24-well polystyrene sterile microplate (Orange Scientific, Braine l’Alleud, Belgium) containing 2 mL of ASW (25‰). These experiments were repeated twice with four replicates per experiment. Plates were covered and sealed with Parafilm^®^ to avoid evaporation and incubated at 25 °C away from any light source for 24 h. Then, the 2 mL of ASW was aspirated from each well and all settled and non-settled cyprids were counted and removed. Wells were then blocked by incubation with 5% BSA in TBST (25 mM Tris-HCl, pH 7.5, 150 mM NaCl, 0.1% Tween 20) for 1 h at room temperature. Wells were then incubated for 1 h at room temperature with the mouse anti-SIPC monoclonal antibody [28] at 1:100 dilution of the cloned cell line culture supernatant in TBST with 5% BSA, washed with TBST (3 × 5 min) and incubated with an HRP-conjugated anti-mouse antibody (1:1000, Cell Signaling Technology, Danvers, MA, USA) for 1 h at room temperature. Wells were then washed with TBST (3 × 5 min) and immunostaining was detected using the 20 × LumiGLO^®^ chemiluminescence kit (Cell Signaling Technology, Danvers, MA, USA) in a Bio-Rad ChemiDoc XRS + Gel Photo Documentation System. To measure the relative intensity of immunoreactive spots on this format of Western blots, images were analyzed with ImageJ 1.52k (NIH, Bethesda, MD, USA), and the adjusted integrated density of the signals was calculated using the formula: Adjusted integrated density (IntDen) = IntDen of experimental region of interest (ROI)—(mean of background ROI × area of experimental ROI). Data were exported and statistically analyzed with GraphPad Prism v.6 using unpaired, two-tailed *t*-tests at *p* = 0.05, with Welch’s correction.

### 4.10. Statistical Analysis

For EC_50_ or IC_50_ (the concentration that results in 50% stimulation or inhibition relative to the control) calculations in GraphPad Prism v.6, all data were fitted to the curve according to the following models for normalized response: *Y* = 100/(1 + 10^((*X* − LogIC_50_))) or *Y* = 100/(1 + 10^((LogEC_50_ − *X*))) depending on the trend of the curve.

## 5. Conclusions

Aiming to identify chemical compounds that can be used as antifouling agents against barnacles, while at the same time testing their ecotoxicity against non-target organisms, we singled out the sesquiterpene perforenol (**9**) as having potent antifouling activity. Perforenol (**9**) displayed settlement-inhibiting activity against *A. amphitrite* cyprids with an IC_50_ value of about 0.5 μM and, most crucially, did not interfere with the settlement seeking behavior of the cyprids of this species. Several other secondary metabolites, isolated from red algae of the genus *Laurencia* and sea hare species of the genus *Aplysia*, had antifouling activity against *A. amphitrite* cyprids but failed to pass our ecotoxicity tests against non-target organisms.

## Figures and Tables

**Figure 1 marinedrugs-17-00646-f001:**
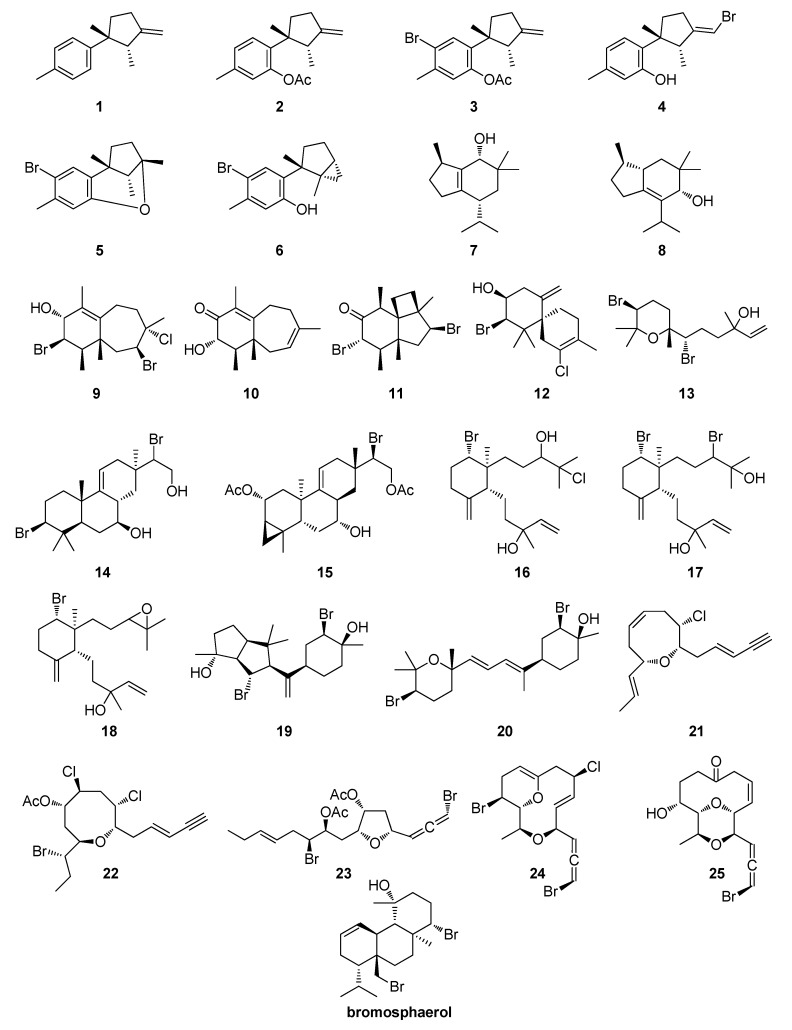
Chemical structures of compounds **1**–**25** isolated from *Laurencia* and *Aplysia* species that were evaluated in the present study and of bromosphaerol that was used as a benchmark.

**Figure 2 marinedrugs-17-00646-f002:**
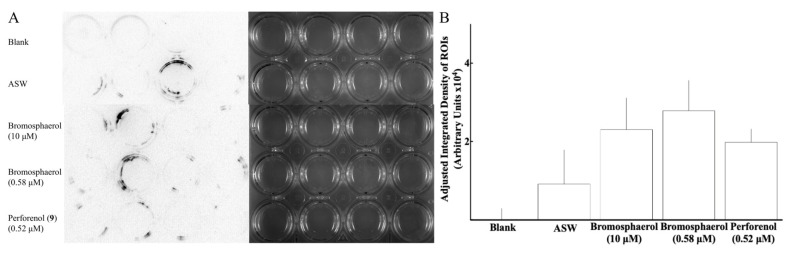
Evaluation of surface exploration activity of *A. amphitrite* cyprids in the presence of perforenol (**9**) and bromosphaerol. (**A**) A mouse anti-SIPC monoclonal antibody that can detect native SIPC deposited by cyprids of *A. amphitrite* on the surface of polystyrene tissue culture plates [28] was used in Western blots after *A. amphitrite* settlement bioassays in the presence of perforenol (**9**) and bromosphaerol or artificial sea water (ASW, serving as control). Sterile tissue culture plates were incubated at 25 °C without (blank) or with 10 cyprids each in 2 mL sterile ASW for 24 h in the presence or absence (ASW) of the indicated concentrations of perforenol (**9**) or bromosphaerol. Solutions and settled and non-settled cyprids were then removed and Western blots were carried out on the wells with a mouse anti-SIPC monoclonal antibody. The bright-field images for each treatment are shown alongside the corresponding immunostained images. A typical image of duplicate experiments is shown. (**B**) The intensity of the detected immunoreactivity in each well is calculated and expressed as adjusted integrated density of the signals at the region of interest (ROI; i.e. each well), after subtraction of the mean value of the blank wells. Results of unpaired, two-tailed t-tests with Welch’s correction, showed that none of the two tested compounds (*p* > 0.05) affected the adjusted integrated density of the detected immunoreactive signals.

**Table 1 marinedrugs-17-00646-t001:** IC_50_, LC_50_ and EC_50_ values (in μM) for settlement, mortality and metamorphosis inhibition of the cyprids of *A. amphitrite* after exposure for 24, 48 and 72 h to the tested secondary metabolites.

Secondary Metabolite	IC_50_ Settlement			LC_50_ Mortality			EC_50_ No Metamorphosis			Therapeutic Ratio (TR = LC_50_/IC_50_)	Settlement/No Metamorphosis Ratio IC_50_/EC_50_
	24 h	48 h	72 h	24 h	48 h	72 h	24 h	48 h	72 h	72 h	72 h
**1**	27.90	14.64	16.21	26.01	229.80	29.46	37.99	ND	ND	1.82	ND
**2**	2.02	18.11	19.77	>1000	790.20	191.20	43.99	38.24	36.47	9.67	0.54
**3**	1.04	1.80	2.00	66.06	382.20	269.20	ND	80.93	109.2	134.60	0.02
**4**	0.12	0.18	0.34	3.55	2.98	2.80	10.35	12.38	5.85	8.24	0.06
**5**	2.26	5.45	12.87	33.14	ND	ND	27.01	0.001	0.287	ND	44.84
**6**	2.42	2.22	1.27	12.31	5.29	2.31	28.33	10.85	14.98	1.82	0.08
**7**	17.49	17.69	10.28	24.75	23.42	24.96	25.41	21.68	53.24	2.43	0.19
**8**	15.83	17.49	6.95	7.76	7.38	5.93	6.81	5.01	5.42	0.85	1.28
**9**	0.520	0.660	0.500	76.23	116.4	49.14	38.17	47.36	57.81	98.28	0.01
**10**	5.24	0.662	4.66	28.83	28.49	28.75	39.29	71.05	57.17	6.17	0.08
**11**	0.003	0.002	0.002	56.14	38.42	ND	0.003	0.002	ND	ND	ND
**12**	0.089	0.073	0.042	0.327	0.194	0.061	0.396	0.250	0.066	1.45	0.64
**13**	1.49	1.06	1.02	29.81	848.7	8.15	46.35	64.68	78.93	7.99	0.01
**14**	0.011	0.065	0.028	1.96	3.61	3.54	6.42	5.15	7.72	126.43	0.00
**15**	2.84	7.02	5.37	31.71	33.35	34.65	63.63	55.09	72.96	6.45	0.07
**16**	1.42	1.96	2.20	>1000	>1000	>1000	58.34	49.09	58.46	>1000	0.04
**17**	0.634	0.806	1.19	ND	ND	>1000	0.057	0.921	100.60	>1000	0.01
**18**	0.046	0.076	0.316	26.24	26.31	20.34	45.36	50.37	48.51	64.37	0.01
**19**	0.400	0.915	0.756	412.9	365.8	573.1	52.80	57.89	70.41	758.07	0.01
**20**	2.57	2.89	3.32	ND	ND	138.3	46.98	11.28	10.05	41.66	0.33
**21**	3.21	7.37	0.575	0.575	1.12	18.63	7.42	18.16	0.001	32.40	575.00
**22**	0.025	0.043	0.055	6.86	10.11	71.34	0.008	0.010	58.66	1297.0	0.00
**23**	0.237	9.18	9.70	ND	ND	ND	0.276	10.31	11.24	ND	0.86
**24** *	ND	9.49	207.50	259.8	527.5	527.5	0.058	63.06	163.4	2.54	1.27
**25** *	62.72	55.62	75.42	643	624.6	618.2	47.07	42.75	53.27	8.20	1.42
Bromosphaerol	0.580	0.624	0.836	17.90	4.57	2.50	0.450	0.267	0.561	2.99	1.49

Secondary metabolites that had a settlement stimulating effect are shown by an asterisk (*). ND: An EC_50_ could not be determined. Detailed graphs of each dose response curve for each secondary metabolite are shown in Appendix A (settlement), Appendix A (mortality) and Appendix A (no metamorphosis). Data for bromosphaerol are shown in Appendix A.

**Table 2 marinedrugs-17-00646-t002:** EC_50_ values (in μΜ) of selected secondary metabolites in various toxicity and cell viability assays.

Secondary Metabolite	*Artemia salina*		*Chaetoceros gracilis*	RTL-W1 Cell Line			HEK293 Cell Line		
	24 h	48 h	96 h	Mitochondria Function	Cell Viability	Lysosome Activity	Mitochondria Function	Cell Viability	**Lysosome** Activity
**4**	21.11	0.739	-	-	-	-	-	-	-
**9**	283.20	189	49.81	>1000	15.97	>1000	10.78	81.10	>1000
**11**	10.05	1.57	4.52	-	-	-	-	-	-
**12**	6.59	0.373	-	-	-	-	-	-	-
**14**	34.19	15.13	1.61	-	-	-	-	-	-
**18**	34.88	38.05	10.71	>1000	0.456	138.2	0.225	>1000	>1000
**19**	31.53	303.8	352.4	>1000	12.81	>1000	0.062	434.10	>1000
**22**	3.07	9.63	2.15	-	-	-	-	-	-
Bromosphaerol	>1000	>1000	4.34	>1000	0.575	143.7	57.61	97.12	340.40

Mitochondria function refers to the Alamar blue/Resazurin assay. Cell viability refers to the Crystal violet assay. Lysosome activity refers to the Neutral red assay. Detailed graphs of each dose response curve for each selected secondary metabolite are shown in Appendix A (*A. salina* toxicity assay), Appendix A (*C. gracilis* toxicity assay), Appendix A (RTL-W1 cell line viability assays) and Appendix A (HEK293 cell line viability assays).

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
