# Peer review of "Evaluation of Antifouling Potential and Ecotoxicity of Secondary Metabolites Derived from Red Algae of the Genus Laurencia"

_marinedrugs, 2019, doi:10.3390/md17110646_

Round 1
Reviewer 1 Report
The authors report on an antifouling study by a cross-species screening process analyzing 25 secondary metabolites isolated from species of the genera Laurencia and Aplysia. Into these substances are include sesquiterpenes, diterpenes and acetogenin class metabolites. Thus, thorough evaluation of the ecotoxicity of selected metabolites against marine arthropods and vertebrate cell lines, perforenol was choosing as most active metabolite. In the screening process bromosphaerol was used as a benchmark.
The work is very well done, with a screening process that allows optimizing the resources and consumption of the different natural metabolites.
Minor comments:
1.- In order to compare the rest of the substances, the structure of bromosphaerol should be included in the main text.
2.- It would be convenient also to include in the manuscript a section of “Conclusions”
Author Response
We would like to thank the Referee for their helpful and insightful comments.
Minor comments:
1) In order to compare the rest of the substances, the structure of bromosphaerol should be included in the main text.
Response:
In response to the referee’s comment we have changed Figure 1 of our manuscript to include the structure of bromosphaerol. In addition, we have added a short text on Page 6, lines 121-122 of the revised manuscript we submit, to further explain the structure of bromosphaerol.
2) It would be convenient also to include in the manuscript a section of “Conclusions”
Response:
In response to the referee’s comment we have included a section of Conclusions on Page 13, lines 395-403 of the revised manuscript we submit.
Reviewer 2 Report
Minor revision.
Introduction, line 54
You wrote “…the herbivorous sea hare Aplysia brasiliana”. For the readers who are not experts in zoology and systematics, there is a first barrier in understanding with what kind of organism you deal in your study. For example, there are numerous marine sponges with very similar scientific name – Aplysina as well as Aplysinella. Furthermore, they are recognized as bromotyrosine-producing organisms. Thus, to overcome this obstacle, please, insert corresponding scientific information about Aplysia as a genus of marine gastropod mollusks.
Line 232: Please insert here the complete scientific name of the mollusc studied as follow:
Aplysia depilans Gmelin, 1791 (Gastropoda: Heterobranchia: Aplysiida: Aplysiidae )
Ref: MolluscaBase (2019). MolluscaBase. Aplysia depilans Gmelin, 1791. Accessed through: World Register of Marine Species at: http://www.marinespecies.org/aphia.php?p=taxdetails&id=138754 on 2019-11-05
You described and discussed brom-containing metabolites, however you have completely overlooked the existence of brominated compounds of sponges origin (see recent paper for overview Kovalchuk et al. (2019) Naturally drug loaded chitin: isolation and applications. Marine Drugs 17:574. Moreover, these compounds have been also studied as antifouling agents See papers by Prof. Peter Proksch group).
Author Response
We would like to thank the Referee for their helpful and insightful comments.
Minor comments:
1) Introduction, line 54. You wrote “…the herbivorous sea hare Aplysia brasiliana”. For the readers who are not experts in zoology and systematics, there is a first barrier in understanding with what kind of organism you deal in your study. For example, there are numerous marine sponges with very similar scientific name – Aplysina as well as Aplysinella. Furthermore, they are recognized as bromotyrosine-producing organisms. Thus, to overcome this obstacle, please, insert corresponding scientific information about Aplysia as a genus of marine gastropod mollusks.
Response:
In response to the referee’s comment we have edited the text on Page 2, lines 54.
2) Line 232: Please insert here the complete scientific name of the mollusc studied as follow:Aplysia depilans Gmelin, 1791 (Gastropoda: Heterobranchia: Aplysiida: Aplysiidae )
Ref: MolluscaBase (2019). MolluscaBase. Aplysia depilans Gmelin, 1791. Accessed through: World Register of Marine Species at:
http://www.marinespecies.org/aphia.php?p=taxdetails&id=138754 on 2019-11-05
Response:
In response to the referee’s comment we have edited the respective text on Page 9, lines 234-235 of the revised manuscript and we have added the respective citation.
3) You described and discussed brom-containing metabolites, however you have completely overlooked the existence of brominated compounds of sponges origin (see recent paper for overview Kovalchuk et al. (2019) Naturally drug loaded chitin: isolation and applications. Marine Drugs 17:574. Moreover, these compounds have been also studied as antifouling agents See papers by Prof. Peter Proksch group).
Response:
We agree with the reviewer that many marine organisms (sponges included) biosynthesize brominated compounds. However, this was not the focus of our study, as we believe it was clearly stated that we evaluated metabolites (not only brominated) of Laurencia origin (including those of dietary origin isolated from herbivorous marine gastropod mollusks of the genus Aplysia).